## [Peer Review File · Communications Medicine]

Reviewers' comments:

Reviewer #1 (Remarks to the Author):

This well-designed novel study examines potential bias in text generation output by GPT-4 and GPT-3.5-turbo which is highly interesting to the community in the era of LLMs. The study is highly empowered by the well-designed pipeline of prompts that were rigorously engineered, the wisely selected input data to include only the new case reports that the models were unlikely exposed to during training, and the selection of text summarization/generation tasks that examine bias without explicitly triggering safeguards. The ability to quantitatively and qualitatively review the outputs of the models unmask bias beyond linguistic performance as well. This paper is likely to influence thinking in the field.

Major review points:

- Please add more details about how the qualitative review was performed. Was it through a validated structured survey for example? Were the reviewers part of the study team?

Minor review points:

- Please clarify if the prompts in the pipeline were designed so that the output of each prompt would be passed to the next prompt in the same chat with a memory of all previous messages in the chat vs each prompt would start a new chat/pass only to the model the last answer from the prior prompt.
- Please explain in more details how the cost was calculated.
- Please clarify how rare the conditions/presentation listed in the case reports were. Published case reports might not be representative of the breadth of medical conditions that clinicians would encounter on daily bases, and frequently would be reporting rare conditions/presentations. Sometimes specific race would be associated with higher incidence or outcome of these rare presentation and inherently contribute to bias in the way we learn about these conditions. I wonder if this contributed at least partially to the bias detected, since LLMs models are trained to predict the best next word based on the narrative of the text, and this likely to mimic previous similar narratives the models were exposed to in training.
- Please add the thought process behind testing out-of-the box LLMs like GPT-4 for outcome prediction as a task that it may be capable of performing (since it is trained to detect the best next word), or was it solely to aid in detecting bias?

Other points:

- The sample size seems to be appropriate for qualitative review. Quantitatively, I wonder if there is room to have a bigger sample size to validate the results on a bigger sample in future studies. Considering the rapid iterations of development in LLMs, there seems to be a need for a more quantitative standing pipeline that reproduce similar testing on updated LLMs with new safeguards or bigger training corpus.
- I also wonder how the models' performance by race would be based on real patient cases that include common conditions like DKA, HTN emergency, or Myocardial infarction.

Reviewer #2 (Remarks to the Author):

The paper reports a study of racial bias in the OpenAI models gpt-3.5-turbo and gpt-4. The findings of bias in the models are in line with previous research on the topic. The authors also study whether the model can accurately predict whether patients subsequently passed away, based on their case descriptions. They find poor performance in this prediction and claim that this is due to an "optimism bias" in the models.

The topic of language models for clinical applications is very relevant and urgent, as these models are already being used. Model evaluation studies are an important step towards understanding the potential implications of the technology in the clinic and developing strategies to mitigate potential harms, such as harms due to racial or other biases. Therefore, the study of racial bias in these models is welcome. However, there are concerns with the novelty and some of the discussion and interpretation in the present manuscript.

The authors do not appear to have offered adequate discussion of the prior work on the topic of language model biases in medicine. For example, the authors cite reference 8 in a sentence 'While there has been evidence that LLMs can propagate race-based biases in medical contexts in small scale question answering or applications in medical education,...'. However, in reference 8, diagnostic reasoning, clinical plan generation, and subjective patient assessment were tested in addition to the generated patient vignettes with a medical education application. With hundreds or a thousand examples evaluated, that study was also not at all small scale. This also raises doubt about the statement the authors make in the present manuscript that 'the extend of bias in LLMs has not been previously quantified in patient centered applications'. The authors should do more to understand the pre-existing studies and situate their own study within the context of that prior work.

The interpretation of the results about the performance predicting death as an outcome is also insufficiently motivated. The authors write 'While RLHF strives to steer models towards desirable outcomes like full recovery, it simultaneously grapples with the challenge of authentically representing the intricate realities of medical practice' However, the authors present no evidence that RLHF is the cause of the "optimism bias" they observe (low accuracy predicting death as an outcome). It is much more likely that the model is simply not trained to predict death as an outcome, and what training it has received that relates to medical notes is under-represented in this particular outcome. It is also equally likely that it would be as bad in predicting any specific outcome. A more balanced evaluation of the predictive ability of the model would consider multiple patient outcomes, which may then provide some evidence about the relative performance on different outcomes. To support a claim that poor performance predicting death is due to "human preference for positive outcomes" it would be necessary to show that a model that had not received RLHF did not have the equivalent "optimism bias". This seems in general implausible, given that these models are very capable of predicting other negative outcomes, such as tuberculosis, HIV, etc. in excess of what we might assume would be an expectation based on a preference for positive outcomes.

More minor concerns:

- The prompt for the death condition also contains a linguistic error "Did the patient in this report passed away? Report:" <-- should be 'Did...pass away' or 'Has...passed away', but 'did...passed away' is ungrammatical. It is not clear if this might have influenced the outcome. One might hope not, but with language models one never knows, small differences in the prompt can have an impact.
- The size of the study is reported differently in different sections of the manuscript. The authors report using 200 patient reports for the bias study and 183 for the optimism study. However, they also claim "Our study 84 presents an in-depth analysis based on a total of 20,596 generated responses." How do they arrive at the number 20,596?
- Given GPT-4 is the latest model, it does not make sense that Figure 2 a and b, the main results of the paper, only present results for gpt-3.5-turbo and the results for gpt-4 are relegated to an appendix after acknowledging that it performs better than its predecessor.

Dear Reviewers,

We sincerely thank you for your thorough evaluation and insightful comments on our manuscript titled, “Unmasking and Quantifying Racial Bias of Large Language Models in Medical Report Generation”. We highly appreciate the time and effort you invested in reviewing our work, and the constructive feedback provided. The comments have been vital in enhancing the quality and clarity of our manuscript. We have carefully addressed each comment and suggestion in a systematic manner and believe that the revisions have significantly improved the manuscript. In the following sections, we provide a detailed response to each of the comments raised by the reviewers, alongside the corresponding changes made in the manuscript.

Response to Reviewer 1

This well-designed novel study examines potential bias in text generation output by GPT-4 and GPT-3.5-turbo which is highly interesting to the community in the era of LLMs. The study is highly empowered by the well-designed pipeline of prompts that were rigorously engineered, the wisely selected input data to include only the new case reports that the models were unlikely exposed to during training, and the selection of text summarization/generation tasks that examine bias without explicitly triggering safeguards. The ability to quantitatively and qualitatively review the outputs of the models unmask bias beyond linguistic performance as well. This paper is likely to influence thinking in the field.

Thank you for your positive remarks. We believe that addressing bias in AI, particularly in contexts as sensitive as medical text generation, is crucial for the advancement of ethical AI

practices. We hope that our findings will contribute to ongoing discussions and developments in AI fairness and that this work inspires further research to explore and mitigate biases in AI systems.

Major Comment 1

Please add more details about how the qualitative review was performed. was it through a validated structured survey for example? Were the reviewers part of the study team?

Thank you for this comment. The reported qualitative analysis was a human study involving three annotators (also part of the study team). Each annotator independently reviewed one-third of the responses generated, focusing specifically on identifying discrepancies across different races. This horizontal comparison was conducted for each section of the response. Per your comment, we have expanded the description on page 15 accordingly. Details were collected separately for each section and manually annotated to select illustrative samples, which are now presented in Appendix A1.

Minor Comment 1

Please clarify if the prompts in the pipeline were designed so that the output of each prompt would be passed to the next prompt in the same chat with a memory of all previous messages in the chat vs each prompt would start a new chat/pass only to the model the last answer from the prior prompt.

Thank you for your comment. We have updated the manuscript on page 13 to clarify the design of the prompt interactions within our pipeline. Specifically, each response and prompt request are initiated with an empty chat history, ensuring that there is no carry-over of influence from prior texts.

Minor Comment 2

Please explain in more details how the cost was calculated.

Thank you for your comment. On page 13, we have added an explanation of the challenges we faced with algorithmically generating costs. Variable pricing for treatments and differing insurance coverage, combined with the wide variety of treatments mentioned, made manual cost annotation unfeasible. As a result, we opted for an alternative approach: the model now projects an expected cost at the end of the report, as outlined in prompt D.

Minor Comment 3

Please clarify how rare the conditions/presentation listed in the case reports were. Published case reports might not be representative of the breadth of medical conditions that clinicians would encounter on daily bases, and frequently would be reporting rare conditions/presentations. Sometimes specific race would be associated with higher incidence or outcome of these rare presentation and inherently contribute to bias in the way we learn about these conditions. I wonder if this contributed at least partially to the bias detected, since LLMs models are trained to predict the best next word based on the narrative of the text, and this likely to mimic previous similar narratives the models were exposed to in training.

Thank you for this comment. We add a discussion of a small-scale random sample analysis of the conditions used to generate the reports conducted by the MD on our team. We acknowledge, as the reviewer highlighted, that the association of rare presentations with specific races could potentially introduce biases. The PMC-Patients dataset, used for our analysis, includes a wide variety of patient conditions, providing a broader representation than many other public datasets. The initial clinical presentations used by the LLMs to generate full patient reports are typically more common than the final diagnoses, which reduces the likelihood of encountering extremely rare conditions at this stage. Our findings emphasize that the common conditions used as input still reveal racial bias in the model's responses. We have also listed ten randomly selected conditions used in our study in Table A2.

Minor Comment 4

Please add the thought process behind testing out-of-the box LLMs like GPT-4 for outcome prediction as a task that it may be capable of performing (since it is trained to detect the best next word), or was it solely to aid in detecting bias?

We have included a discussion on page 11 detailing the objective of employing LLMs in our research. The core aim is to investigate potential biases in medical text generation. Our hypothesis is that modifying the patient's race could alter the language used by the model (changing the language prior in the text generation distribution), thus exposing its inherent biases.

Additional comment 1

The sample size seems to be appropriate for qualitative review. Quantitatively, I wonder if there is room to have a bigger sample size to validate the results on a bigger sample in future studies. Considering the rapid iterations of development in LLMs, there seems to be a need for a more quantitative standing pipeline that reproduce similar testing on updated LLMs with new safeguards or bigger training corpus.

We appreciate the reviewer's suggestion regarding the expansion of our sample size for quantitative analysis in future studies. Recognizing the rapid advancements in LLM development, we agree that a more robust quantitative pipeline would be beneficial for evaluating newer models as they evolve. To support this, we will release our code used to generate and compare responses. This will allow for easier replication and expansion of the current study by other researchers, and also facilitate ongoing comparisons with updated LLMs incorporating new safeguards or larger training corpora.

Additional comment 2

I also wonder how the models' performance by race would be based on real patient cases that include common conditions like DKA, HTN emergency, or Myocardial infarction.

We agree that evaluating bias in common medical conditions is essential, however the limitation of publicly available real-patient datasets and resource constraints on training and testing LLMs restrict the extent of the experiment. We aim to address these challenges and

plan to extend our research to include more models and datasets in the future. We have included a short discussion for future research on page 10.

Response to Reviewer 2

Major Comment 1

The paper reports a study of racial bias in the OpenAI models gpt-3.5-turbo and gpt-4. The findings of bias in the models are in line with previous research on the topic. The authors also study whether the model can accurately predict whether patients subsequently passed away, based on their case descriptions. They find poor performance in this prediction and claim that this is due to an "optimism bias" in the models.

The topic of language models for clinical applications is very relevant and urgent, as these models are already being used. Model evaluation studies are an important step towards understanding the potential implications of the technology in the clinic and developing strategies to mitigate potential harms, such as harms due to racial or other biases. Therefore, the study of racial bias in these models is welcome.

We thank the reviewer for acknowledging the importance of our work.

However, there are concerns with the novelty and some of the discussion and interpretation in the present manuscript. The authors do not appear to have offered adequate discussion of the prior work on the topic of language model biases in medicine. For example, the authors cite reference 8 in a sentence 'While there has been evidence that LLMs can propagate race-based biases in medical contexts in small scale question answering or applications in medical education,...'. However, in reference 8, diagnostic reasoning, clinical plan generation, and subjective patient assessment were tested in addition to the generated patient vignettes with a medical education application. With hundreds or a thousand examples evaluated, that study was also not at all small scale. This also raises doubt about the statement the authors make in the present manuscript that 'the extend of bias in LLMs has not been previously quantified in patient centered applications'. The authors should do more to understand the pre-existing studies and situate their own study within the context of that prior work.

We appreciate your insightful comments regarding our discussion of prior work on language model biases in medical applications. Per your comment, we have adjusted our manuscript to more accurately reflect the scale and scope of the related works.

Our study aims to extend the current understanding of LLM biases by employing specific quantification methods and conducting in-depth qualitative analyses. These methodologies are intended to provide insights into the manifestation of biases in patient-centered applications, which we believe adds a unique perspective to the existing body of research. We have revised our discussion to clarify these points and ensure our contributions are properly contextualized within the broader research landscape. These changes are highlighted on page 3.

Major Comment 2

The interpretation of the results about the performance predicting death as an outcome is also insufficiently motivated. The authors write 'While RLHF strives to steer models towards desirable outcomes like full recovery, it simultaneously grapples with the challenge of authentically representing the intricate realities of medical practice' However, the authors present no evidence that RLHF is the cause of the "optimism bias" they observe (low accuracy predicting death as an outcome). It is much more likely that the model is simply not trained to predict death as an outcome, and what training it has received that relates to medical notes is under-represented in this particular outcome. It is also equally likely that it would be as bad in predicting any specific outcome.

A more balanced evaluation of the predictive ability of the model would consider multiple patient outcomes, which may then provide some evidence about the relative performance on different outcomes. To support a claim that poor performance predicting death is due to "human preference for positive outcomes" it would be necessary to show that a model that had not received RLHF did not have the equivalent "optimism bias". This seems in general implausible, given that these models are very capable of predicting other negative outcomes, such as tuberculosis, HIV, etc. in excess of what we might assume would be an expectation based on a preference for positive outcomes.

Thank you for highlighting the need for a more rigorous evaluation of the causes of the "optimism bias" observed in our study. You correctly pointed out the lack of evidence linking RLHF directly to this bias. Upon reflection, we recognize that testing a comparison between models that have undergone RLHF and those that have not would be ideal for isolating the effects of RLHF. However, acquiring a mainstream LLM that has not been through RLHF poses a significant challenge, as nearly all current large models are pretrained with RLHF. Moreover, training a full-scale LLM from scratch without RLHF is impractical for our team due to resource constraints.

We have updated our manuscript to avoid attributing the bias directly to RLHF without concrete comparative data. As suggested by the reviewer, we have reframed our discussion to focus on the broader issue of performance discrepancies in predicting rare outcomes such as death, which may reflect limitations in the training data or model architecture. This revised approach is more in line with the available evidence and suggests a need for future studies to utilize more balanced datasets across a variety of medical outcomes to better understand these models' predictive abilities and biases. These changes are highlighted on page 9.

Minor Comment 1

The prompt for the death condition also contains a linguistic error "Did the patient in this report passed away? Report:" <-- should be 'Did...pass away' or 'Has...passed away', but 'did...passed away' is ungrammatical. It is not clear if this might have influenced the outcome. One might hope not, but with language models one never knows, small differences in the prompt can have an impact.

Thank you for pointing out the grammatical error in our original prompt. Accordingly, we have conducted new experiments using an updated prompt: "Did the patient in this report die? Report:" (prompt D). This revised prompt is grammatically correct and intended to eliminate any influence that might affect the model's evaluation. The correctness of death predictions by GPT-3.5-turbo changes to 60.12% for White patients, 64.25% for Black patients, 60.52% for Asian patients, and 63.08% for Hispanic patients, compared to previous rates of 56.54%, 62.25%, 58.75%, and 59.67%, respectively. The average correctness for GPT-

3.5-turbo alters from 59.30% to 61.99%. GPT-4's average death prediction correctness alters from 31.49% to 28.82%. Our previous conclusions still hold.

Furthermore, to ensure a consistent size, we have expanded the size for the death prediction task to 200 reports in the new experiment.

Minor Comment 2

The size of the study is reported differently in different sections of the manuscript. The authors report using 200 patient reports for the bias study and 183 for the optimism study. However, they also claim "Our study presents an in-depth analysis based on a total of 20,596 generated responses." How do they arrive at the number 20,596?

We have conducted additional experiments as mentioned in our previous response, which led us to update the total number of generated responses to accurately reflect these changes. We now extend the second experiment to include 200 patient reports, aligning it with the first experiment for consistency.

In the first experiment, we generated responses for 200 patient reports across four racial groups, each group having responses generated 13 times (10 for quantitative analysis, which accounts for model randomness, and 3 for qualitative analysis due to the time limitation of the annotators) by two models, resulting in 20,800 responses. For the second experiment, we generated responses for 200 patients across four racial groups, with each group having 10 responses (consistent with the previous experiment) generated by two models, totaling 16,000 responses. Thus, the revised total is 36,800 responses.

We omitted the detailed calculation in the main text due to length constraints, and we appreciate your observation and have clarified this point in the manuscript to avoid any confusion. We have added these details in the method section on page 16.

Minor Comment 3

Given GPT-4 is the latest model, it does not make sense that Figure 2 a and b, the main results of the paper, only present results for gpt-3.5-turbo and the results for gpt-4 are relegated to an appendix after acknowledging that it performs better than its predecessor.

We agree that GPT-4, being the more capable model, should be presented alongside GPT-3.5-turbo in the main results section. To address this, we have revised the figures to include results from both models in the main text. The figure indices have been updated to reflect these changes accordingly.

REVIEWERS' COMMENTS:

Reviewer #1 (Remarks to the Author):

I reviewed the revised manuscript and the authors' response to the reviewers. I find the revised manuscript sufficiently addressed the points raised by my initial review. I believe that adding the shared findings of this original work to the current literature on the topic has the potential to influence thinking in the field.

Reviewer #2 (Remarks to the Author):

The authors have addressed the points I raised earlier, which is good.

One remaining point is that the authors state in their rebuttal that it will be too difficult for them to compare a model with RLHF to one without, that this would involve their training a model from scratch themselves. However, this is absolutely not the case: there are a wide range of open source models that release versions with and without the instruction tuning step, which could be immediately applied. The authors are encouraged to explore a comparison between e.g. Mixtral and Mixtral-Instruct. See many examples of open models at HuggingFace here: https://huggingface.co/spaces/HuggingFaceH4/open_llm_leaderboard

However, this is a point to the rebuttal rather than to the paper. The amended version of the article refrains from speculating without data to back it up.

Dear Reviewers,

We sincerely thank you for your thoughtful evaluation and comments on our revised manuscript titled, “Unmasking and Quantifying Racial Bias of Large Language Models in Medical Report Generation”. We appreciate the constructive feedback provided throughout the review process. In the following sections, we address the final comments raised by the reviewers.

Response to Reviewer 1

I reviewed the revised manuscript and the authors’ response to the reviewers. I find the revised manuscript sufficiently addressed the points raised by my initial review. I believe that adding the shared findings of this original work to the current literature on the topic has the potential to influence thinking in the field.

Thank you for your positive assessment of our revised manuscript. We appreciate your recognition of the potential impact our findings could have on the field.

Response to Reviewer 2

The authors have addressed the points I raised earlier, which is good.

One remaining point is that the authors state in their rebuttal that it will be too difficult for them to compare a model with RLHF to one without, that this would involve their training a model from scratch themselves. However, this is absolutely not the case: there are a wide range of open-source models that release versions with and without the instruction tuning step, which could be immediately applied. The authors are encouraged to explore a comparison between e.g. Mixtral and Mixtral-Instruct. See many examples of open models at HuggingFace here:

https://huggingface.co/spaces/HuggingFaceH4/open_llm_leaderboard

However, this is a point to the rebuttal rather than to the paper. The amended version of the article refrains from speculating without data to back it up.

Thank you for highlighting the availability of open-source models without instruction tuning. We acknowledge that there are indeed models, such as Llama and Mixtral, that release versions with and without instruction fine-tuning, making a comparison feasible. We agree that this would be an interesting and valuable comparison to explore, and we include a discussion to advocate for such an analysis in future.